# The Issue of Land Subsidence in Coastal and Alluvial Plains: A Bibliometric Review

Carla Buffardi *[ID] and Daniela Ruberti [ID]

Department of Engineering, University of Campania L. Vanvitelli, Via Roma, 9, 81031 Aversa, Italy
* Correspondence: carla.buffardi@unicampania.it

**Abstract:** Land subsidence (LS) is becoming one of the major problems in coastal and delta cities worldwide. Understanding the current LS situation and the research trends is of paramount importance for further studies and addressing future international research networks. We analyzed the LS-related literature available from the Scopus database. The use of a single database avoided the redundancy of articles, while excluding some subject areas was useful to obtain only studies related to LS. By using VOSviewer and CiteSpace tools, we conducted a bibliometric analysis by considering title, keywords, and abstract to identify the temporal development, the geographical origin, and the area of study of the research. The results revealed a considerable heterogeneity of approaches, thematics, study areas, and research output trends. China, the US, and Italy are the major contributors to the scientific production, but the higher number of articles is not always related to the extension of the LS phenomenon in these countries. The monitoring approach differs worldwide, and univocal modeling is still lacking; from the analysis of the keywords, it is clear that the focus of most studies is on the relationship with the hydrological/hydrogeological aspects. Since the 2000s, however, the development of SAR technologies has boosted the study of the phenomenon from a different point of view.

**Keywords:** land subsidence; review; bibliometrics; geohazards; coastal-alluvial plains; research methods

## 1. Introduction

Subsidence is the gradual or sudden sinking of the ground surface [1] due to the consolidation of sediments causing subsurface movement of earth materials as a result of increasing effective stress. Land subsidence (LS) can have different natural drivers, including tectonics, isostatic adjustment, and the spatial and temporal variability of sediment compaction [2,3], but it can also have anthropic causes like groundwater pumping [4] or mining [5]. This phenomenon can cause damage to the environment and consequently to human life and represents a major concern from different points of view, involving socioeconomic, environmental, and protection issues [6].

Although LS affects different geographic settings, it represents a crucial problem in alluvial-coastal and delta cities [7–9]. For the future, it is considered a bigger issue because of the rising of sea level (SLR) [10], especially for those areas that are sinking faster than the sea level is rising. In these settings, the combined effect of LS and SLR can affect marine-coastal environments, contributing to the loss of coastal lowlands, can lead to the salinization of the aquifer, the destruction of infrastructure, and increased vulnerability to flooding [2,11]. The estimated economic damage associated with the phenomenon worldwide is billions of dollars annually [10]. In the last century, for example, subsidence caused by groundwater depletion affected 200 different locations in 34 countries, and it is estimated to affect an area of 2.2 million km$^2$ in 2040 and 19% of the global population [12].

To varying degrees, all countries are affected by LS, although the problem is underestimated and poorly investigated in many locations.

Among the United States of America, for example, subsidence greatly affects the Mississippi delta, with very variable rates, not only because of the huge extent of the area but

also because of the different triggers. The age and the characteristics of the sediments result in displacement rates from 2 mm/y up to 16 mm/y [5,13]; in industrial areas or where there is a massive extraction of material from the subsoil, subsidence rates can reach 70 mm/y [5]. However, there are still no strategies aimed at mitigating this phenomenon, as underground resource extraction is a very important economic resource for the whole country [14].

On the Asian continent, the first LS event was recorded in 1964 in Shanghai (China), and in the following decades, the phenomenon was detected in other regions of the country. In the last decades, the average subsidence rate is 70 mm/y [15], affecting an area of 49,000 km$^2$ and causing damage amounting to 100 million dollars [1]. According to different studies, these land displacements are mostly due to anthropogenic causes, in particular, the extraction of fluids from the subsoil, like in Shangai or Henan Province [1,16–18], while, in the southern part of the country, around the city of Nannin, the main cause has been found in the composition of the subsoil [19].

Other nations are also affected by LS, including India, Indonesia, and Japan, where the population has been facing damage caused by the sinking of the ground for decades [20–23]. In Indonesia, subsidence rates range between 2 and 24 cm/y and affect at least 6 different cities. The capital Jakarta is severely affected by rates that reach up to 17.5 cm/y. The main causes, in this case, are groundwater depletion and the natural compaction of the layers in the subsoil, while in the coastal area of Semarang, for example, the main cause is attributed to the expansion of urbanization. In Japan, the first evidence of subsidence was recorded in Tokyo and dates back to 1910, while in 1968, the rate of vertical displacement was 24 cm/y, undoubtedly caused by the large volume of fluids and gases extracted from poorly compacted layers in the subsurface [22,24].

The Netherlands is one of the European countries most heavily impacted by the phenomenon. The main drivers are considered to be both the weight of buildings and other infrastructure causing consolidation of the layers present in the subsoil [25] and the combination of primary compaction, creep, and oxidation of peat [26]. In Italy, subsidence has been known since the second half of the 20th century, in particular in the Po Valley area, where the main triggers are the compaction of the less cohesive layers of the subsoil [27,28] and the extraction of natural resources [29–32]. More recently, the phenomenon was also reported in another coastal area in southern Italy [33–35].

Many other countries are affected by LS, and remarkable efforts have been made to assess its causes. Several studies addressed different important topics on LS, including methods to measure ground displacement; monitoring and remediation techniques; and forecasting the phenomenon to prevent damage to the environment and human life. Few thoughtful works of literature refer to global-scale issues [12,36,37], while most of the research to date deals with local problems, at a country scale [1,38–40] or smaller areas [41–46]; other studies have developed analytical methods for specific triggers [4,47,48].

Here we present a review of the available peer-reviewed literature based on bibliometric analysis, with the aim to highlight the main topics of the issue, the trends, and approaches of the research, providing a timeline of the research on a global scale [49]. The idea is to highlight the strengths and weaknesses of the research, the collaborations at an international level, and the lack of information, with the aim of stimulating awareness of this phenomenon and more effective cooperation between countries.

## 2. Materials and Methods

### 2.1. Data Collection

The research considered any literature containing the expression "Land Subsidence" in the title, keywords, or abstract in the SCOPUS database, which is commonly used in academic research [50–52].

The number of publications obtained from the SCOPUS database was reduced using the classification carried out by the website itself in subject areas that allowed to exclude from the analysis the documents not relevant to our investigation as they did not refer

to LS but to phenomena of a different nature, often in the biomedical field, along with immunology, dentistry, pharmacology, and molecular biology (Table 1).

**Table 1.** Lists of different subject areas included and excluded from the analysis.

| Subject Areas Included | Subject Areas Excluded |
| --- | --- |
| Agricultural and Biological Sciences | Arts and Humanities |
| Chemistry | Biochemistry, Genetics and Molecular Biology |
| Computer Science | Business, Management and Accounting |
| Decision Sciences | Chemical Engineering |
| Earth and Planetary Sciences | Dentistry |
| Energy | Economics, Econometrics and Finance |
| Engineering | Immunology |
| Environmental Science | Medicine |
| Mathematics | Multidisciplinary |
| Physics and Astronomy | Pharmacology, Toxicology and Pharmaceutic |
| Social Sciences | |

Further screening was based on the type of publication. Only articles, conference papers, reviews, and book chapters in the English language were considered (Figure 1). The database was last updated on December 2022 and the final number obtained is 7140 publications, covering the period between 1881 and 2022 (Table S1). The information obtained is about the temporal development of the search, the geographical origin, and the area of study of the research, and finally the analysis of the keywords. Data were also obtained by analyzing abstracts.

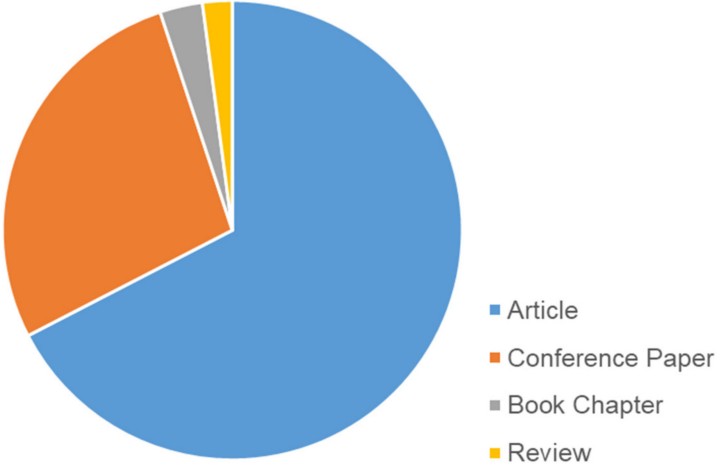

**Figure 1.** Distribution of the main types of publications used in the bibliometric analysis.

*2.2. Data Elaboration*

The whole database was downloaded from Scopus. Given the large amount and variability of data obtained, it was necessary to process them according to conceptual schemes and make use of specific software, according to the workflow in Figure 2.

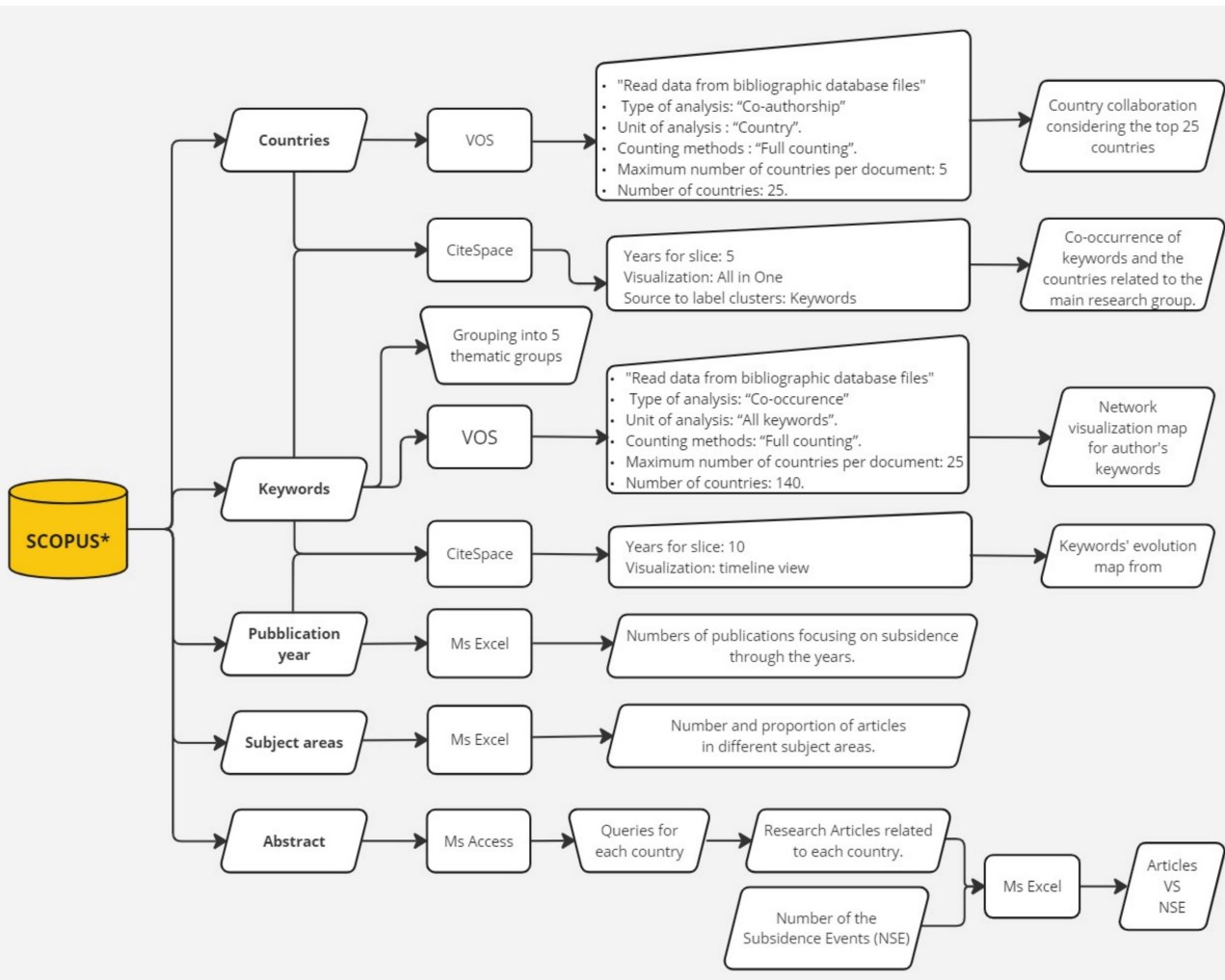

**Figure 2.** Flowchart of the used methodology. The symbol (*) indicates the filtering performed on the database upstream, according to Table 1.

The entire database was uploaded into Ms Access, including the abstracts; by using different queries, data on articles that focused on each available study area were obtained.

Through the use of Ms Excel, it was possible to analyze the growth over the years of the studies and the different subject areas to which the publications belonged, using and adapting the different graphs available.

Further elaborations were made using specific software dedicated to the processing of bibliometric data, using different settings according to output: (i) VOSviewer_1-6-17 was used for analyses requiring only one input datum; this was the case for the analysis concerning the collaboration between countries and the co-occurrence of keywords; (ii) CiteSpace 6.2.R2 (Advanced) was used for the analyses involving the combination of two different data inputs, which allowed the cross-check of different data and the final plot of the results. Using this software, the analyses concerning the temporal development of the keywords and the relationship between the different states and the main keywords were carried out.

## 3. Results and Discussion

### 3.1. Publication Outputs: General Trends and Subject Area

The oldest article in the database is written by Garnder in 1881 [53] and refers to a previous paper entitled "Subsidence and Elevation, and on the Permanence of Oceans" [54], which is not on Scopus and is mainly focused on the ground deformation of the ocean's

surface. Until the 1950s, only 11 publications were recorded, and their main topics are mainly related to the development of delta areas [55,56]; after this period, the scientific production concerning subsidence gradually increased, except for a few peaks of increased production in 1986 and 1995.

A significant increase in research is recorded around 2010 (Figure 3); the total number of publications up to 2009 is 2249, while 4152 documents were published between 2010 and 2022, i.e., 2 times as many.

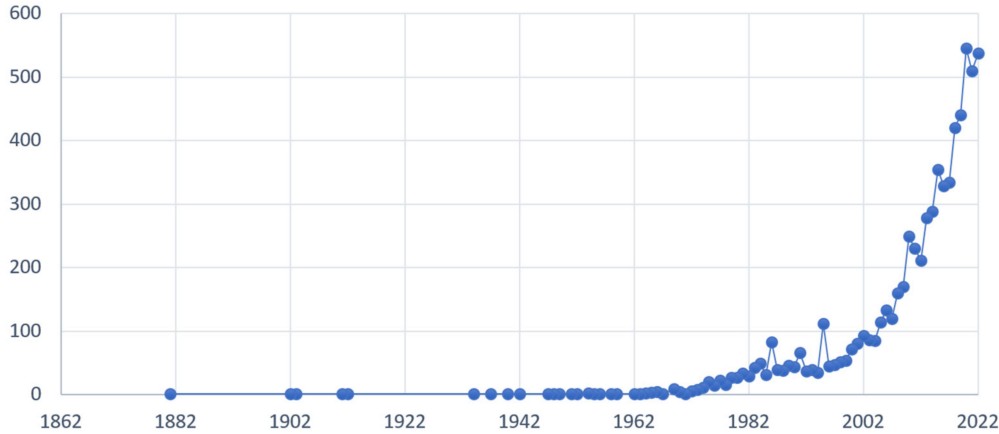

**Figure 3.** Number of publications focusing on subsidence through the years.

The explanation of this increase can be associated with the greater possibility of highlighting the phenomenon of LS and monitoring it over time thanks to the intensification of studies concerning the analysis of the satellite data; indeed, as we can see in Figure 4, the production of related work increased substantially between 2008 and 2009, almost 35%.

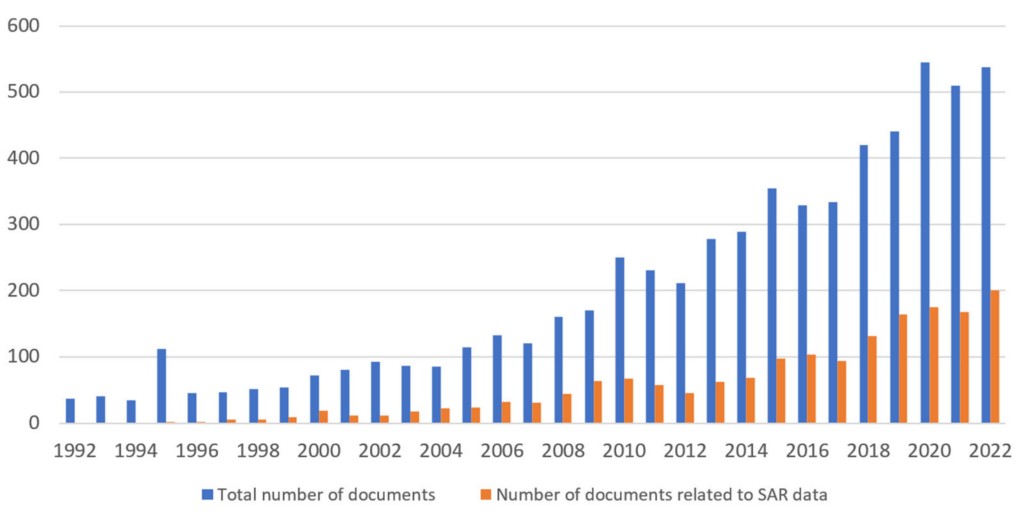

**Figure 4.** Comparison of the total published documents on subsidence and those focusing on the use of satellite-based methods.

However, it is noteworthy to specify that studies related to satellite analysis first appeared in the 1990s [3,57,58]: the pioneering paper by Gabriel et al., in 1989, introduced Synthetic Aperture Radar to measure the small movement of the ground with good resolution and was based on SAR interferometry used to measure large patches of land. Alongside these factors, it should also be considered that LS has become the major concern of coastal areas and deltas [59], especially since the population has grown enormously in these environmental contexts. Also, the consequences of the phenomenon in these areas have received the attention of many scientists in the last 10–15 years [60–63], highlighting

not only the contribution of technological innovations, but also the need to constantly monitor the development of the phenomenon.

Regarding the thematic areas of publication of the considered articles, it should be emphasized that the Scopus classification consists of 334 subject areas that are grouped into macro-categories (Figure 5). Each article is usually assigned to a subject area by the editorial board of the journal in which it is published, and therefore the same research topic can be present in different categories [64].

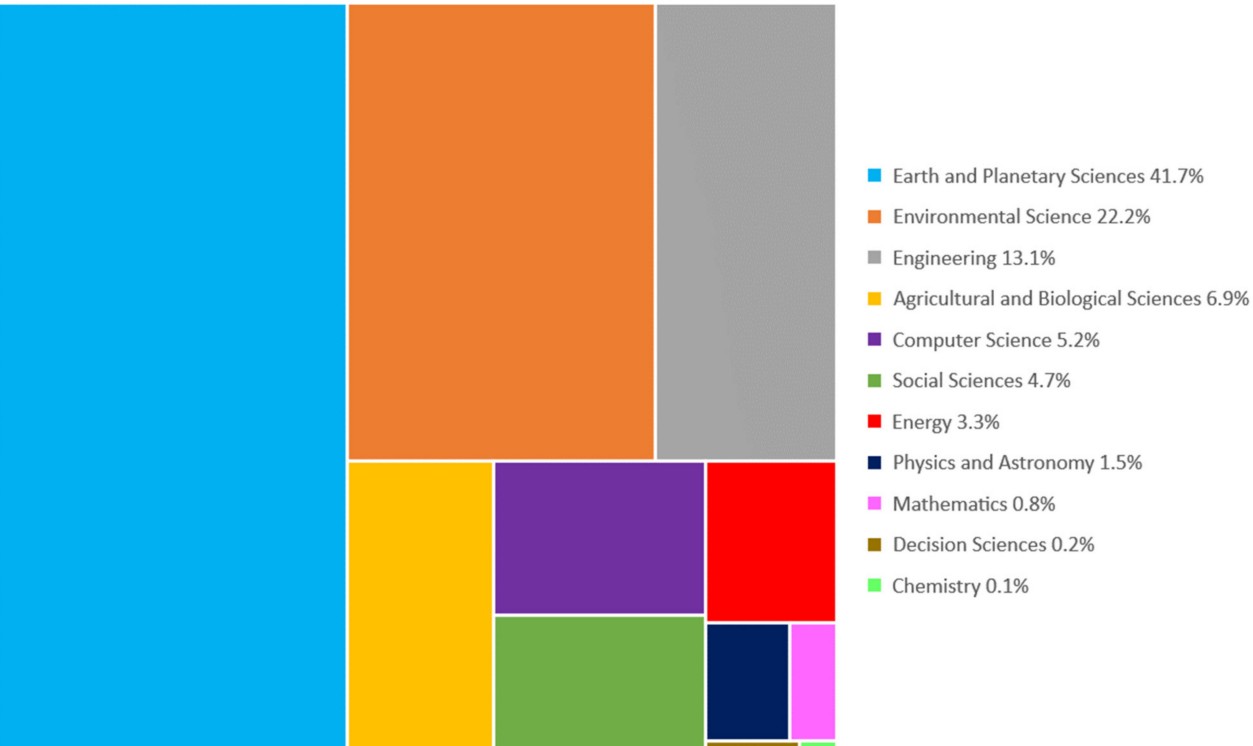

**Figure 5.** Number and proportion of articles in different subject areas.

The subject areas considered in this review are different, but they are often interconnected and mutually contained. For instance, more than 50% of the publications are related to the area "Earth and Planetary Sciences", which appears too broad and heterogeneous in content; in many articles, subsidence is only a marginal aspect of research based, for example, on climate change [65,66] or the development of delta areas [67,68]. Among the other subject areas, "Environmental science" includes many contributions, mainly related to the hydrological aspects, as does "Engineering", which is associated with the modeling of subsidence through mathematical methods [18,69], the use of satellite technologies to measure displacement [70], or the composition of the subsoil [71–73]. Many articles related to mining activities or fluid withdrawal were published under "Energy".

As can be seen in Figure 5, the phenomenon appears to be very transversal, also falling within the area of: (i)"Agricultural and biological sciences", in relation to the land use or the groundwater issues, which are subsidence-inducing phenomena; (ii) "Social Sciences", linked to the consequences on the economy of the regions affected by LS and related phenomena, such as, for example, the flooding of coastal areas [74–76]; and (iii) "Computer Sciences", mainly related to the analysis of SAR data and the modeling of the phenomenon.

This variety of contributions in different fields underlines how much the phenomenon requires a strongly multidisciplinary approach to study, both in scientific terms (modeling) and in terms of risk assessment and economic management.

### 3.2. Global Trends and Collaborations in LS Studies

A collaboration network analysis was performed to highlight the countries involved in research on LS, considering the number of publications and institutional collaborations (Figure 6).

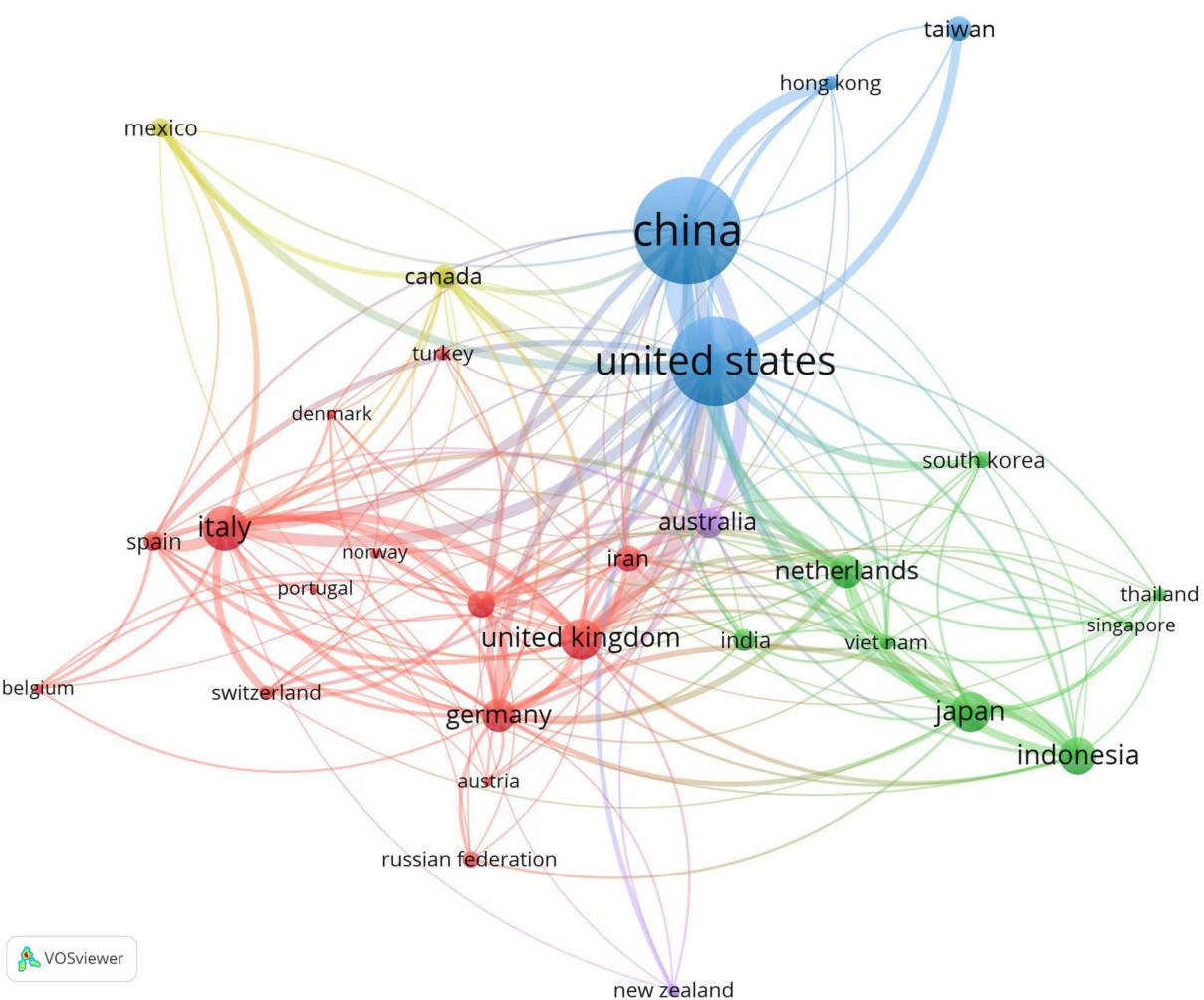

**Figure 6.** Network map for country collaboration based on the number of publications and considering the top 25 countries. The node size and thickness of the connecting lines are proportional to the number of documents assigned to each country. The connections represent the collaboration network among research institutions.

China and the United States are the leading countries that have contributed significantly to the LS research with respectively 1704 and 1278 publications, followed by Italy with 435. The sum of these articles represents 53% of the total number of publications. The great interest in this issue in these countries can be attributed to different factors: about 50% of Chinese cities are experiencing land subsidence, according to a 2016 study by the China Geological Survey, and some of them are subsiding more than 1 m; in California's Central Valley, subsidence alone, due to abundant groundwater extraction, caused economic damage estimated at over USD 2.7 billion between 2012 and 2015 [77]. These countries are followed by the United Kingdom, Japan, Indonesia, and the Netherlands; as mentioned in the introduction, the last three are among the countries most affected by subsidence, unlike the United Kingdom.

The connection lines in Figure 6 show the collaboration network, which defines the degree of communication between various countries and their institutions [78]. The thickness of the connection lines underlines the strength of the cooperation. Again, the US and China appear to be

the core countries of this network, with the US having more extensive international cooperation. Some European countries (i.e., Belgium, Switzerland, Norway, Denmark, among others), on one side, and others from tropical regions (such as Thailand, South Korea, or New Zealand) have few publications and weak international collaborations.

The different colors in Figure 6 highlight three main clusters among the considered countries [79]. The main cluster (shown in blue) consists of China and the United States, as already discussed above. The second one (red) is mainly formed by European countries, with a significant contribution from Italy, France, the UK, and Germany, and some Asian countries. The latter populated almost all of the third cluster (green) together with The Netherlands. One minor cluster (violet) has its core in New Zealand and shows itself as well-distanced from the other clusters with few publications and poor connections.

On the whole, there is no clear demarcation between all of the clusters, indicating that there is cross-country collaboration on LS research, which allows us to analyze different aspects of a phenomenon that affects the areas of study, promoting innovation and sharing different knowledge. In this respect, it is noteworthy to underline that the scientific productivity of a country is not proportional to the LS events recorded in the country itself. The consequence of this is that the country of origin of the articles is not always strictly related to the area of study.

To better understand this aspect, an analysis of the abstracts was carried out, extracting the countries of interest in each article. In Figure 7, the output of this analysis is compared with the number of subsidence events (NSEs) taken by Herrera-García et al., (2021) [12]. In many cases, the number of publications is not related to the number of subsidence events: an NSE of 16 in China corresponds to 450 articles, while in the case of Spain, an NSE of 10 corresponds to just 18 documents. Thus, even though the number of NSEs is very comparable, the research, in this case, is very limited, highlighting a lack of awareness of the phenomenon compared to other countries. Another interesting case is given by the United States: although the number of abstracts is considerable, it is not proportional to the elevated NSE, especially when it is compared to China. In this case, the reasons for this huge production can be found in the inner policies of these nations, such as high population density and significant financial funds allocated to the analysis of the effects of the phenomenon rather than to the extent of the phenomenon itself in a given area.

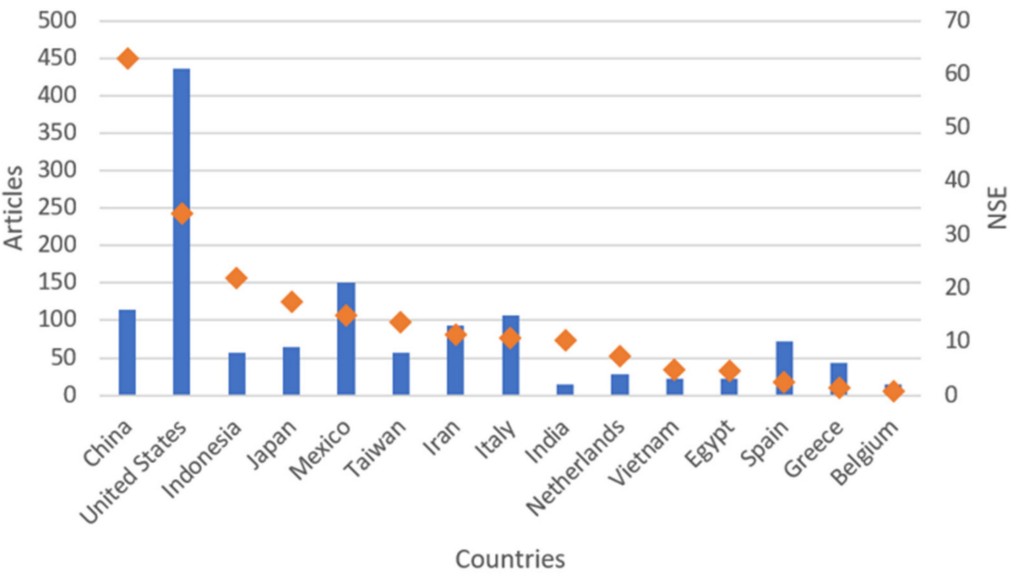

**Figure 7.** Relationship between the number of subsidence events (NSE) and the number of research articles related to each country.

The analysis of the abstracts also revealed that only a few papers deal with the economic aspect, giving rough estimates, or not taking into account all of the damage caused by the phenomenon. The review by Kok and Costa [80] highlights the lack of a common methodology to determine the economic damage of subsidence, also due to the wide variability of the causes of the phenomenon. In many countries, in fact, there are a lot of subsiding areas, and in many cases, the causes are different from each other and different in magnitude, as already explained in the introduction. On this topic, refer to the accurate review provided by Bagheri-Gavkosh [9].

### 3.3. Analysis of Authors' Keywords

Keywords are very important in a research output, as they provide important insights into the content of each publication [49], the most relevant topics, and the main research trends [81]. Only keywords that occurred a minimum of 70 times were used. A total of 143 items were displayed, with 27,148 co-occurrences. They were divided into 5 thematic groups (Table S2), excluding all keywords related to the name of the country or locality (Figure 8).

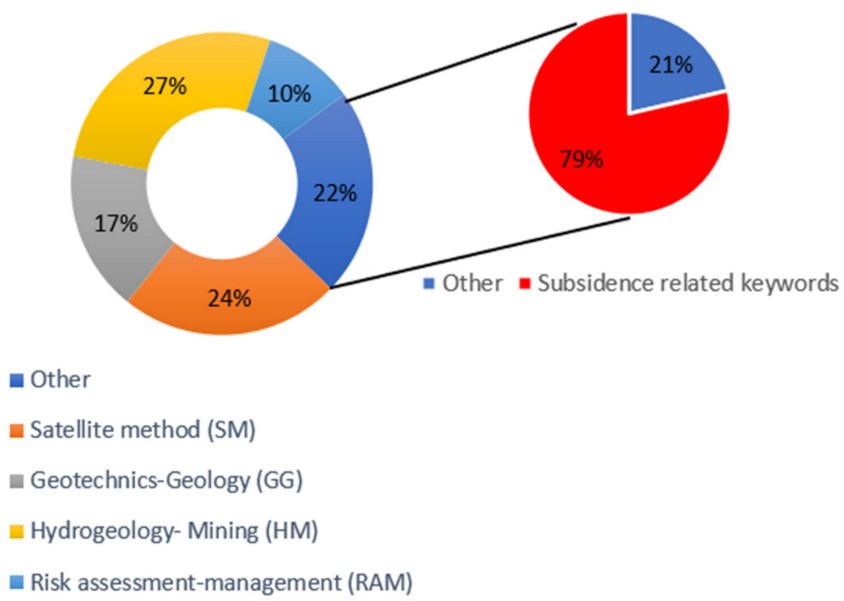

**Figure 8.** Main topics covered by the studies focusing on LS. Five thematic groups of keywords were identified and are reported with their relative percentage of occurrence. The complete list is provided in Table S2. Keywords including the term "subsidence" were included in the "Other" group since they characterize the phenomenon in general; their weight within that group is shown on the right.

Four groups include items with specific reference to (i) Satellite Methods (SM), (ii) Geotechnics-Geology (GG), (iii) Hydrogeology-Mining (HM), and (iv) Risk Assessment-Management (RAM). All keywords that were not strictly related to the previous, were grouped into an "Other" category. The keywords *subsidence* and *land subsidence* were included in the latter group since they characterize the phenomenon in general and the analysis aimed to highlight the relationship between the phenomenon itself and the study approaches. The group with the greatest co-occurrence of keywords is that relating to HM, with 27%, demonstrating that most of the studies address the origin of the phenomenon above all to extraction and mining activity. SM follows, with 24%, although the frequency of these words is documented largely from the end of the 20th century, thanks to the development of satellite surveying techniques, as previously mentioned. The inclusion of the terms *subsidence* and *land subsidence* brings the Other group to 22%, but these two keywords represent about 79% of the group itself. The contribution of the geological and

geotechnical component (GG group), also transversal to SM and HM, is on the order of 17%, while RAM includes 10%.

### 3.3.1. Trajectories of Study

To highlight the trajectories of the studies, the co-occurrence among all keywords identified was examined, choosing to represent those that occurred at least 140 times in a network visualization map (Figure 9).

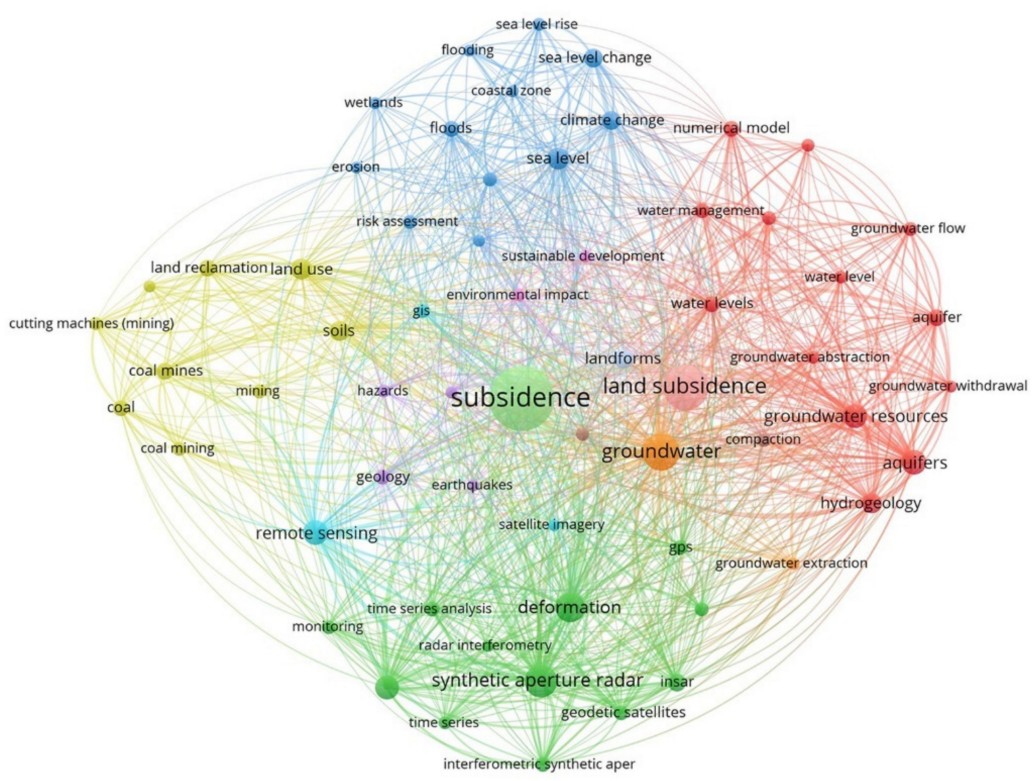

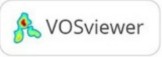

**Figure 9.** Network visualization map of authors' keywords. The node size and thickness of the connecting lines are proportional to the number of documents in which the keyword appears. The colors indicate the cluster the item belongs to, and the connection line between nodes represents co-occurrence; the shorter the distance between the different nodes, the stronger the relationship between the keywords.

Excluding *subsidence* and *land subsidence*, the most used keyword is *groundwater* and other terms related to groundwater resources, highlighting the importance of the relationship between the phenomenon and the variations in groundwater levels, which can be caused by both human activity [82,83] and different characteristics of the subsoil [84,85]. The hydrogeological aspects undoubtedly represent one of the most relevant topics in the research trends highlighted by the bibliometric analysis and which embraces both the reference to the extraction of water from the subsoil (red cluster in Figure 9) and that relating to mining activities (yellow cluster). Great attention is also paid by research to land use and reclamation, which determine strong impacts both on the surface water supply and on relations with the subsoil. The red and yellow clusters in Figure 9 largely coincide with the HM keywords. In this type of representation, however, the shorter the distance between the different nodes is, the stronger is the relationship between the keywords, and the position of the latter is not always in the cluster of those of the same thematic group.

Another important group of keywords is the one related to satellite methods to study and monitor the phenomenon of ground deformation. *Synthetic aperture radar (SAR)*, *inSAR*,

*interferometry*, and *deformation* are the most frequently recurring keywords, among others. They represent specific useful tools to monitor large areas and provide digital elevation models (DEMs) [86–88]. The green cluster contains the most used items in this field and shows good relationships, especially with the keywords that refer to hydrogeology.

The blue cluster, which contains all phenomena connected and/or caused by subsidence, is also quite dense. They range from the connection with *sea level change* and the *climate changes* that influence them to the combined effects of LS and sea level change, such as *floods* or *erosion*, up to the environments that are most affected by the negative effects of LS, including *wetlands* and *coastal zone*. As previously discussed, the coastal sector has received greater attention only in more recent times, both due to the problems deriving from the rising of the sea level due to climate change, and thanks to the possibility of monitoring the phenomenon through satellite techniques. However, the interplay of subsidence, coastal erosion and flooding, and groundwater salinization has pushed the international scientific community to pay increasing attention, above all to the economic damage they involve.

3.3.2. Trends in the Research through Time

As stated above, through the keyword analysis, it is possible to obtain different data along with the trends in the research through time. This type of evaluation allows us to better understand when certain topics became more important than others; this can be observed in Figure 10, where the size of the text is proportional to the frequency of the keywords. Here again, the connection lines refer to co-occurrence.

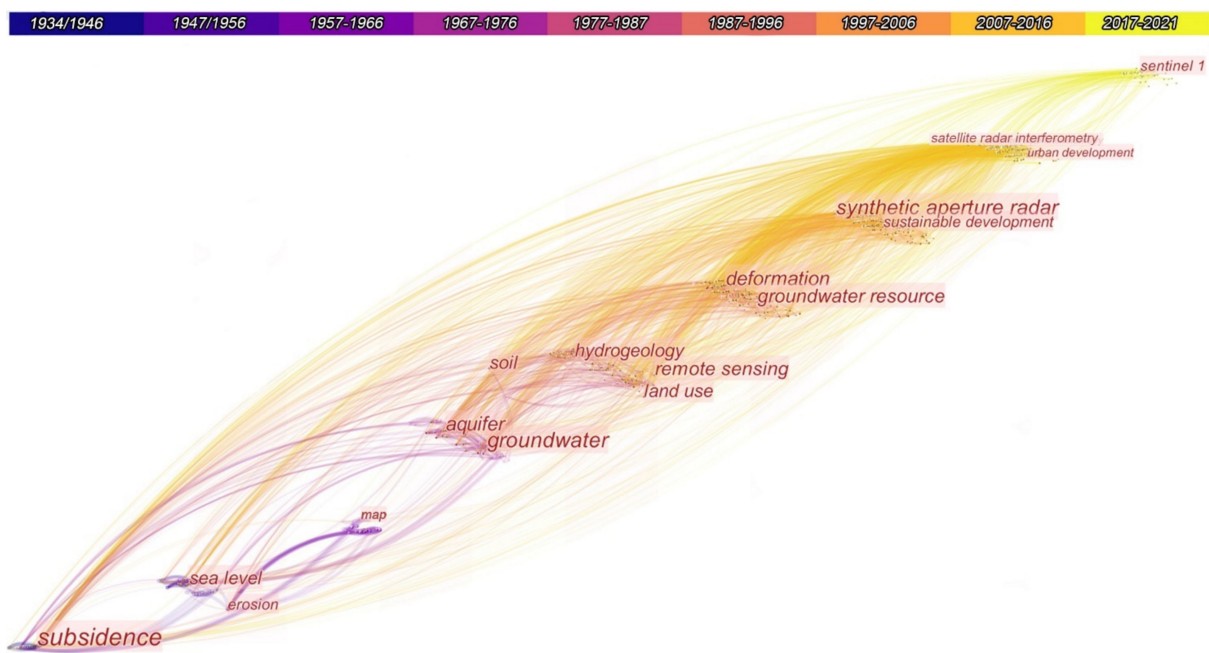

**Figure 10.** Keywords' evolution map from 1934–2021 displayed in time slices shown at the top boundary. The text size represents the relative frequencies of keyword co-occurrence.

Between 1934 and 1943, the main keyword obviously is *subsidence*; in the following decade, the phenomenon is associated with variations in sea level, in order to understand the relationship between these two events [89] and the problem of coastal erosion, which is seen as evidence of the occurrence of subsidence [90,91]. The main keyword during the decade from 1957 to 1966 is *map*, replaced in the following years by words relating to the *aquifer*, especially its variation, whether due to extraction [92] or compaction [93]. After this time, the focus shifts to understanding the effect of land use on LS [94,95] and to the first applications of remote sensing to measure the phenomenon [96]. This represents the beginning of a new phase of

studies on evolving technologies, which from the 1990s onwards began to dominate subsidence research, sometimes accompanied by themes such as sustainable development, in particular on sustainable groundwater management [97–99] or urban development, to study the connection between urbanization and ground sinking [100–102].

### 3.3.3. Distribution of Research Power

A network visualization map is provided to highlight the main trends in research in different countries (Figure 11). Within the selection made for the keywords, 23 countries were highlighted among those that occurred a minimum of 70 times. They represent 74% of all those present in the analyzed database. The nodes represented with circles indicate the countries, while rhombuses indicate the keywords. The different colors highlight clusters. The connection lines identify the main focus of the research in the different countries of each cluster.

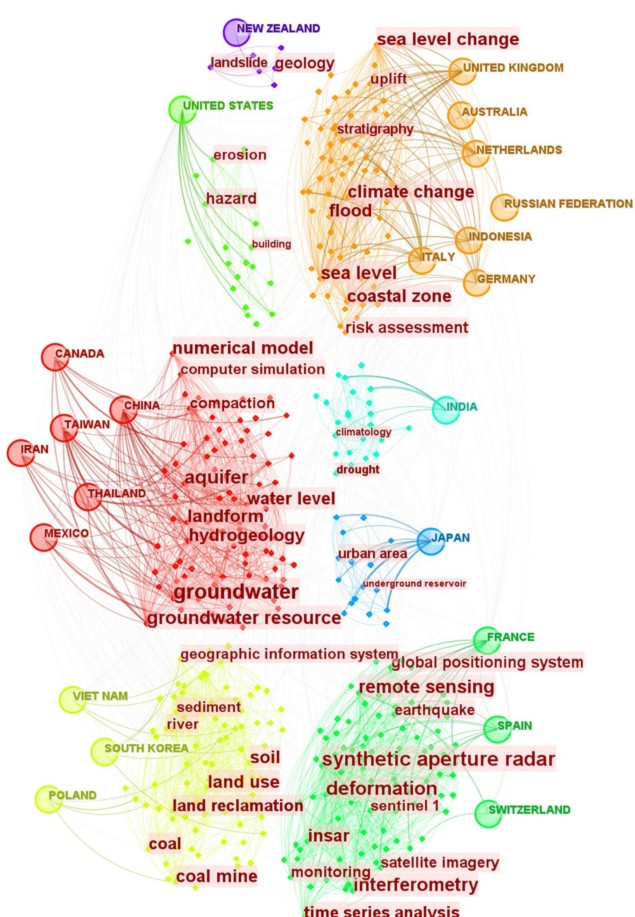

**Figure 11.** Network visualization map for co-occurrence of keywords and the countries related to the main research groups.

The orange cluster includes Australia, Germany, Indonesia, Italy, Netherlands, Russia, and the UK. It is the largest cluster and includes the themes relative to the risks related to subsidence [103], like flooding [104,105], but also to the main causes such as sea level rise [106,107] and climate change. Studies in Canada, China, Iran, Mexico, Taiwan, and Thailand (red cluster) are mainly related to hydrogeology [108–110], with a lesser extent to mitigation measures related to the exploitation from the subsoil [83,111–113]. The same countries are involved in the numerical modeling of the phenomenon [114,115] and the elaboration of the satellite data [116–118]. Poland, South Korea, and Vietnam (yellow cluster) focus their research mainly on the effects of mining and land use changes [21,119,120]. The analysis of satellite-related data is the main focus of the research in France, Spain, and Switzerland (green cluster): almost all keywords linked to these countries

refer to SAR, interferometry, or remote sensing monitoring tools [121–123]. Four countries seem to form separate clusters. The US, although characterized by a high incidence of the LS phenomenon and the core country of a large network of research centers, focuses its attention above all on the risks associated with erosion caused by subsidence [124–126]; fewer connections are highlighted with the red cluster. New Zealand, very distant from other research groups and characterized by scarce scientific production, mainly deals with issues related to the geology and the composition of the substrate [127–129]. India, which shows weak international collaborations, mainly focuses on the effect of climate change and drought [130,131]. Lastly, the main attention of the studies in Japan is concentrated on urban areas and their growth, which are seen both as a cause of the phenomenon and the places affected mainly by LS [132,133], especially in coastal areas [134].

As can be seen from the figure, however, the demarcation between the various clusters is never clear-cut.

The data analyzed do not allow us to identify exactly the main causes of subsidence in the various countries, as these are not always indicated in the keywords and it is often difficult to derive them from the abstracts. However, taking into account what was reported in previous elaborations (cf. Figures 9 and 11), integrating the abstracts with literature data, along with the review by Bagheri-Gavkosh [9], a synthetic scheme is provided which highlights, for each country, the main issues addressed in the literature, assuming that they actually correspond to the causes (Table 2).

**Table 2.** Lists of different causes for each country analyzed.

| Countries | Natural | Layer Compaction/Consolidation | Tectonic | Fault | Groundwater Extraction | Mining | Oil/Gas Extraction | Land Use Change |
|---|---|---|---|---|---|---|---|---|
| Australia | | X | X | X | X | | | |
| Canada | X | | | | | | X | |
| China | | | | | X | | | |
| France | | | | | X | X | X | |
| Germany | | | | X | | X | | |
| India | | | | X | X | | X | |
| Indonesia | X | | | | X | | | X |
| Italy | X | X | X | | X | X | X | X |
| Iran | | X | | | X | | | |
| Japan | | X | | | X | | X | |
| Mexico | X | X | X | X | X | X | X | X |
| New Zeland | X | | | X | X | | X | |
| Poland | X | X | X | | X | X | X | X |
| Russia | X | X | X | X | | X | X | |
| South Korea | | | X | | X | X | | |
| Spain | X | X | X | | X | | | |
| Swizerland | | | | X | X | | | |
| Taiwan | X | | | | | X | | |
| Thailandia | X | | | | X | | | |
| The Netherlands | X | | | | X | | X | |
| United Kingdom | X | X | | | X | X | | |
| United States | X | X | X | X | X | X | X | X |
| Vietnam | | X | X | | X | X | X | |

Table 2 takes into consideration the natural causes, differentiating those of sedimentary origin and those of tectonic origin, and then the causes of anthropic origin, differentiating

those linked to extractions of fluids from the subsoil and those deriving from land use changes.

On the whole, the main causes refer to extraction activities, whether they are fluid withdrawal (water, oil, gas) or mining; these are reported in almost all of the countries considered. The major contributions to their modeling come mainly from Spain and the Netherlands, where the phenomenon is widely characterized. Fewer references instead refer to natural causes such as, for example, compaction and/or consolidation phenomena. Finally, in some regions the influence of tectonics is dominant, as in Switzerland or Germany.

It is clear that this output is far from giving a precise picture of the main causes of subsidence since it is based on more than 7000 input data; however, it highlights, even more, the complexity of the phenomenon and the need to identify guidelines aimed at the "implementation and evaluation of risk assessments and mitigation measures, and the definition of resource-management strategies that support sustainable development in areas vulnerable to land-level lowering" (https://www.landsubsidence-unesco.org/ accessed on 30 April 2023).

## 4. Conclusions

The bibliometric analysis of LS research using VOSviewer and CiteSpace enabled us to identify the methods commonly used to study the LS that affects most of the alluvial-coastal and delta areas in the world. The results revealed a considerable heterogeneity of approach, thematic variables, study areas, and research output trends.

The contributions of different countries to the LS issue were evaluated. China, the United States of America, and Italy contributed 53% of the total number of LS-related articles. However, this scientific production is not proportional to the subsidence events recorded in the country itself. This probably indicates less awareness and/or attention to the phenomenon on the part of other countries. Furthermore, the output of the collaborations shows a strong influence of some countries in international collaborations even if the phenomenon is not intense in those countries.

The number of publications about LS increased in the 20th century with the improvement of SAR technologies, which has given a significant boost to the study of the phenomenon by providing powerful tools for assessing the phenomenon over very large areas and continuously over time. This is evidenced by the strong occurrence among keywords of terms referring to these technologies starting from these years.

The keyword co-occurrence was in fact used to clarify the core literature, background of knowledge, and research branches. Assessment of causes related to fluid extraction covers approximately 27% of research topics. On the other side, satellite-related techniques cover almost 24% of the research trajectories. They form the main branch of research, which relies on hydrogeology and numerical modeling of satellite data. The geological and geotechnical aspects of the soil and those relating to the assessment and management of risks related to subsidence are transversal to the previous research topics, covering respectively 17% and 10%. This highlights the extension of LS research into earth sciences, engineering, physics, and environmental sciences.

Ultimately, LS is a worldwide problem enhanced by economic development. Coastal and deltaic countries around the world are likely to suffer this hazard that, coupled with climate change, induced sea level rise. The output of this bibliometric study highlights that the LS phenomenon has not been monitored with the same extension worldwide, and a thorough and univocal modeling is still lacking. Moreover, some countries have formed strong international networks with very high numbers of research publications; it is hoped that this will encourage researchers to develop and create larger networks of connection with other countries and researchers to develop a more incisive impulse to research and help mitigate the risks associated with LS.

**Supplementary Materials:** The following supporting information can be downloaded at: https://www.mdpi.com/article/10.3390/rs15092409/s1, Table S1: Database containing the articles used for the analysis; Table S2: Keyword categorization.

**Author Contributions:** Conceptualization, C.B. and D.R.; methodology, C.B.; software, C.B.; validation, C.B. and D.R.; data curation, C.B.; writing—original draft preparation, C.B. and D.R.; writing—review and editing, C.B. and D.R.; supervision, D.R. All authors have read and agreed to the published version of the manuscript.

**Funding:** The scholarship and research activities of Carla Buffardi were part of the Environmental, Design and Innovation Ph.D. program. This research was also funded by the SEND intra-university project, financed by the "V:ALERE 2019" funds (VAnviteLli pEr la RicErca) by the University of Campania "L. Vanvitelli" (Grant ID: B68D19001880005).

**Data Availability Statement:** Publicly available datasets were analyzed in this study. These data can be found here: https://www.scopus.com/ accessed on 4 April 2023.

**Acknowledgments:** The authors kindly acknowledge the insightful comments by the three anonymous reviewers who greatly helped to improve the manuscript.

**Conflicts of Interest:** The authors declare no conflict of interest.

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
