# Peer review of "The Issue of Land Subsidence in Coastal and Alluvial Plains: A Bibliometric Review"

_remotesensing, doi:10.3390/rs15092409_

Round 1
Reviewer 1 Report
The manuscript titled "The issue of land subsidence in coastal-alluvial plains: a review", submitted by Buffardi and Ruberti is within the journal's scope. The manuscript is focused on a much broader issue of land subsidence in coastal-alluvial plains.
The title can be modified to indicate which type of review the authors are performing (here bibliometric)
The authors are advised to use high-resolution figures (Figures 1 to 4). The present quality of the figures is not good.
The authors should update the review till 2023.
In lines 54-56 "Other nations are also affected by LS, along with India, Indonesia and Japan, where the population has been facing the damage caused by the sinking of the ground for decades." authors should cite references.
Authors can take help of the following for strengthening the introduction:-
https://www.sciencedirect.com/science/article/pii/S0048969721012602
The author should highlight the need of the study, i.e. why this review is needed.
Author Response
Dear Reviewer,
we kindly acknowledge your insightful comments. The figures and the manuscript were modified accordingly. Replies to specific comments are listed below.
The manuscript titled "The issue of land subsidence in coastal-alluvial plains: a review", submitted by Buffardi and Ruberti is within the journal's scope. The manuscript is focused on a much broader issue of land subsidence in coastal-alluvial plains.
The title can be modified to indicate which type of review the authors are performing (here bibliometric)
Answer: We changed the title to specify the type of review
- The authors are advised to use high-resolution figures (Figures 1 to 4). The present quality of the figures is not good.
Answer: Figures were upgraded
- The authors should update the review till 2023.
Answer: The review has been updated till December 2022. It is not significant to include publications from the first two months of 2023 since the manuscript was submitted in early 2023.
- In lines 54-56 "Other nations are also affected by LS, along with India, Indonesia and Japan, where the population has been facing the damage caused by the sinking of the ground for decades." authors should cite references.
Answer: references were added
- Authors can take help of the following for strengthening the introduction:-
https://www.sciencedirect.com/science/article/pii/S0048969721012602
Answer: we used the suggested references to update the introduction
- The author should highlight the need of the study, i.e. why this review is needed.
Answer: We specified in the introduction that the bibliometric study of the subsidence literature aims to highlight the main topics of the issue, the trends, and the approaches of the research, providing a timeline of the research on a global scale. Unlike a bibliographic review, this one highlights all aspects related to the study of subsidence in various countries. By the output of this bibliometrics, it was highlighted that the LS phenomenon had not been monitored with the same extension worldwide and thorough and univocal modeling is still lacking. This can open the way for new strands of collaboration or approaches to study
Reviewer 2 Report
Make a separate table mentioning the causes of subsidence reported by literature from different countries. Also include in the discussion, regional differences in causal factors of subsidence reported in various studies. For example, in the North Himalayan region, subsidence and upliftment are going on due to tectonic movement mainly, but in the North-West Plains of India, subsidence is caused by groundwater depletion. Such discussion at the global level will be excellent for better understanding and definitely be appreciated.
Beside of comparing number of subsidence event and number of publications, authors must include the estimated (or approximated) economic damage due to subsidence and then relate with number of publications. It is because economic damages cause social concern and put in the limelight for awareness and further encourages to conduct research activities.
The author can also add simple statics on how much spending a country is doing on R&D and relate it with the number of publications. It would explain why the research outcomes vary in different countries.
I also suggest categorising the publications into a few classes based on the Impact factor and check the majority of publication is going towards with category of impact factor journal.
In Figure 6, use a composite bar graph instead of a bar and line graph, because a line graph is used to depict any temporal pattern, and here the subjective information is presented through the figure.
Figure 7 image quality needs to be improved as it can't be viewed clearly.
Author Response
Dear Reviewer,
we kindly acknowledge your insightful comments. The figures and the manuscript were modified accordingly. Replies to specific comments are listed below.
- Make a separate table mentioning the causes of subsidence reported by literature from different countries. Also include in the discussion, regional differences in causal factors of subsidence reported in various studies. For example, in the North Himalayan region, subsidence and upliftment are going on due to tectonic movement mainly, but in the North-West Plains of India, subsidence is caused by groundwater depletion. Such discussion at the global level will be excellent for better understanding and definitely be appreciated.
Answer: Some details were discussed in the introduction. Based on what was said in the introduction, the bibliometric analysis made it possible to analyze the incidence of the phenomenon in individual countries on the basis of what is indicated in the documents available in Scopus (i.e. keywords, title, abstract). From these data, it is very difficult to discriminate what is required by the reviewer because it is not always indicated. Carrying out the analysis for only a few literature data in which this information is specified would make the result incomplete and fragmentary.
- Beside of comparing number of subsidence event and number of publications, authors must include the estimated (or approximated) economic damage due to subsidence and then relate with number of publications. It is because economic damages cause social concern and put in the limelight for awareness and further encourages to conduct research activities.
The author can also add simple statics on how much spending a country is doing on R&D and relate it with the number of publications. It would explain why the research outcomes vary in different countries.
Answer: most publications do not mention data on economic damages. The analysis of keywords and abstracts did not show references to these study parameters. Therefore it is not possible to add this evaluation. A comment has been added that highlights precisely this lack and the need to make economic evaluations for the future.
- I also suggest categorising the publications into a few classes based on the Impact factor and check the majority of publication is going towards with category of impact factor journal.
Answer: Some reviews also focus the bibliometric analysis on the impact factor of the journals. We believe that this analysis does not make much sense as the introduction of this parameter dates back to the beginning of 2000 while our analysis goes back to 1880. Furthermore, the IF of the journals changes over the years and it would be necessary to reconstruct it referring to the year of publication of each article. Therefore the work would be long and not significant.
- In Figure 6, use a composite bar graph instead of a bar and line graph, because a line graph is used to depict any temporal pattern, and here the subjective information is presented through the figure.
Answer:. The comment is reasonable. However, the analysis program does not allow the output of a graph with composite bar graphs and different reference scales for each scale bar. For this reason, we deleted the connecting lines and left only the points.
- Figure 7 image quality needs to be improved as it can't be viewed clearly.
Answer: Done.
Reviewer 3 Report
(1)Land subsidence is not only in coastal-alluvial plains , but also in the inland area where groundwater exploitation or mining are very serious. As a summary article, this obviously cannot cover all situations. I hope the author can supplement this part.
(2) What format (accdb) is the file in the attachment? Many readers do not know how to open it.
Author Response
Dear Reviewer,
we kindly acknowledge your insightful comments. The figures and the manuscript were modified accordingly. Replies to specific comments are listed below.
- Land subsidence is not only in coastal-alluvial plains , but also in the inland area where groundwater exploitation or mining are very serious. As a summary article, this obviously cannot cover all situations. I hope the author can supplement this part.
Answer: To tell the truth, the revision also covers more internal areas and for this reason the title has been slightly modified. The emphasis on coastal alluvial areas is related to the fact that subsidence makes them extremely vulnerable to other phenomena such as saline intrusion or coastal erosion.
- What format (accdb) is the file in the attachment? Many readers do not know how to open it.
Answer: We changed the format of the file. It is easily readable.
Reviewer 4 Report
Since I did not have much time for reading the paper, please use this comment only as a general opinion on the quality of the paper not like a review.
In In my opinion, this paper is an impressive review aimed at collecting a lot of interesting data from the existing international literature on subsidence. While I don't consider myself an expert in this field of research, I guess this paper is well written, the data are well organized in the text and in the figures, and the conclusions are well supported. As a result of my first impression, I think this review paper can be accepted in its present form.
Kind regards
Author Response
Dear Reviewer,
Thank you for your appreciation.
Round 2
Reviewer 1 Report
Dear Authors
Thanks for addressing the previous comments.
I would like to mention that It is easy for researchers who know how to use the tools mentioned in this study to generate such results as shown in the paper. No effort is required to answer the stated research questions. Please add a step-by-step process for the search and analysis of the data.
The analysis of this paper is too descriptive. Most of the analyses shown in this paper are fairly easy to create.
Further, there is a lack of discussion of the results of this study. Therefore authors must improve the discussion section which is too generalized and simple.
Author Response
Dear Reviewer,
we kindly acknowledge your insightful comments. The figures and the manuscript were modified accordingly. Replies to specific comments are listed below.
- I would like to mention that It is easy for researchers who know how to use the tools mentioned in this study to generate such results as shown in the paper. No effort is required to answer the stated research questions. Please add a step-by-step process for the search and analysis of the data.
Answer: The Methodology has been improved and a flowchart added to the text to explain the step-by-step analysis.
- The analysis of this paper is too descriptive. Most of the analyses shown in this paper are fairly easy to create.
Answer: We have better explained how the analyses were performed. The database, downloaded from Scopus, was not analysed as it was but through some processes and filtering based on specific reasoning, explained in the text. The documents considered are more than 7000 and for each one the title, keywords and abstract have been considered. It is not an automatic analysis.
- Further, there is a lack of discussion of the results of this study. Therefore authors must improve the discussion section which is too generalized and simple.
Answer: The Discussion section was improved.
Reviewer 2 Report
In introduction section, may be in the last paragraph, authors must mention why subsidence in coastal and alluvial plains is crucial to review. Mountainous and Mining areas are often affected by LS activities much more than plain areas.
In discussion section, it is obligatory to include what are the different factors responsible for the LS in various area. Otherwise, this review becomes incomplete.
Author Response
Dear Reviewer,
we kindly acknowledge your insightful comments. The figures and the manuscript were modified accordingly. Replies to specific comments are listed below.
- In introduction section, may be in the last paragraph, authors must mention why subsidence in coastal and alluvial plains is crucial to review. Mountainous and Mining areas are often affected by LS activities much more than plain areas.
In the Introduction section, as well as in the Results and Discussion one, it has been stressed the reason why subsidence in coastal and alluvial plains is crucial to review
In discussion section, it is obligatory to include what are the different factors responsible for the LS in various area. Otherwise, this review becomes incomplete.
Done